# User-Interactive Offline Reinforcement Learning

**Phillip Swazinna**
Siemens & TU Munich
Munich, Germany
swazinna@in.tum.de

**Steffen Udluft**
Siemens Technology
Munich, Germany
steffen.udluft@siemens.com

**Thomas Runkler**
Siemens & TU Munich
Munich, Germany
thomas.runkler@siemens.com

## Abstract

Offline reinforcement learning algorithms still lack trust in practice due to the risk that the learned policy performs worse than the original policy that generated the dataset or behaves in an unexpected way that is unfamiliar to the user. At the same time, offline RL algorithms are not able to tune their most important hyperparameter - the proximity of the learned policy to the original policy. We propose an algorithm that allows the user to tune this hyperparameter at runtime, thereby addressing both of the above mentioned issues simultaneously. This allows users to start with the original behavior and grant successively greater deviation, as well as stopping at any time when the policy deteriorates or the behavior is too far from the familiar one.

## 1 Introduction

Recently, offline reinforcement learning (RL) methods have shown that it is possible to learn effective policies from a static pre-collected dataset instead of directly interacting with the environment (Laroche et al., 2019; Fujimoto et al., 2019; Yu et al., 2020; Swazinna et al., 2021b). Since direct interaction is in practice usually very costly, these techniques have alleviated a large obstacle on the path of applying reinforcement learning techniques in real world problems.

A major issue that these algorithms still face is tuning their most important hyperparameter: The proximity to the original policy. Virtually all algorithms tackling the offline setting have such a hyperparameter, and it is obviously hard to tune, since no interaction with the real environment is permitted until final deployment. Practitioners thus risk being overly conservative (resulting in no improvement) or overly progressive (risking worse performing policies) in their choice.

Additionally, one of the arguably largest obstacles on the path to deployment of RL trained policies in most industrial control problems is that (offline) RL algorithms ignore the presence of domain experts, who can be seen as users of the final product - the policy. Instead, most algorithms today can be seen as trying to make human practitioners obsolete. We argue that it is important to provide these users with a utility - something that makes them want to use RL solutions. Other research fields, such as machine learning for medical diagnoses, have already established the idea that domain experts are crucially important to solve the task and complement human users in various ways Babbar et al. (2022); Cai et al. (2019); De-Arteaga et al. (2021); Fard & Pineau (2011); Tang et al. (2020). We see our work in line with these and other researchers (Shneiderman, 2020; Schmidt et al., 2021), who suggest that the next generation of AI systems needs to adopt a user-centered approach and develop systems that behave more like an intelligent tool, combining both high levels of human control and high levels of automation. We seek to develop an offline RL method that does just that. Furthermore, we see giving control to the user as a requirement that may in the future be much more enforced when regulations regarding AI systems become more strict: The EU's high level expert group on AI has already recognized "human autonomy and oversight" as a key requirement for trustworthy AI in their Ethics Guidelines for Trustworthy AI (Smuha, 2019). In the future, solutions found with RL might thus be required by law to exhibit features that enable more human control.

In this paper, we thus propose a simple method to provide users with more control over how an offline RL policy will behave after deployment. The algorithm that we develop trains a conditional policy, that can after training adapt the trade-off between proximity to the data generating policy on

the one hand and estimated performance on the other. Close proximity to a known solution naturally facilitates trust, enabling conservative users to choose behavior they are more inclined to confidently deploy. That way, users may benefit from the automation provided by offline RL (users don't need to handcraft controllers, possibly even interactively choose actions) yet still remain in control as they can e.g. make the policy move to a more conservative or more liberal trade-off. We show how such an algorithm can be designed, as well as compare its performance with a variety of offline RL baselines and show that a user can achieve state of the art performance with it. Furthermore, we show that our method has advantages over simpler approaches like training many policies with diverse hyperparameters. Finally, since we train a policy conditional on one of the most important hyperparameters in offline RL, we show how a user could potentially use it to tune this hyperparameter. In many cases of our evaluations, this works almost regret-free, since we observe that the performance as a function of the hyperparameter is mostly a smooth function.

## 2 RELATED WORK

**Offline RL** Recently, a plethora of methods has been published that learn policies from static datasets. Early works, such as FQI and NFQ (Ernst et al., 2005; Riedmiller, 2005), were termed batch instead of offline since they didn't explicitly address issue that the data collection cannot be influenced. Instead, similarly to other batch methods (Depeweg et al., 2016; Hein et al., 2018; Kaiser et al., 2020), they assumed a uniform random data collection that made generalization to the real environment simpler.

Among the first to explicitly address the limitations in the offline setting under unknown data collection were SPIBB(-DQN) (Laroche et al., 2019) in the discrete and BCQ (Fujimoto et al., 2019) in the continuous actions case. Many works with different focuses followed: Some treat discrete MDPs and come with provable bounds on the performance at least with a certain probability Thomas et al. (2015); Nadjahi et al. (2019), however many more focused on the continuous setting: EMaQ, BEAR, BRAC, ABM, various DICE based methods, REM, PEBL, PSEC-TD-0, CQL, IQL, BAIL, CRR, COIL, O-RAAC, OPAL, TD3+BC, and RvS (Ghasemipour et al., 2021; Kumar et al., 2019; Wu et al., 2019; Siegel et al., 2020; Nachum et al., 2019; Zhang et al., 2020; Agarwal et al., 2020; Smit et al., 2021; Pavse et al., 2020; Kumar et al., 2020; Kostrikov et al., 2021; Chen et al., 2019; Wang et al., 2020; Liu et al., 2021; Urpí et al., 2021; Ajay et al., 2020; Brandfonbrener et al., 2021; Emmons et al., 2021) are just a few of the proposed model-free methods over the last few years. Additionally, many model-based as well as hybrid approaches have been proposed, such as MOPO, MOReL, MOOSE, COMBO, RAMBO, and WSBC (Yu et al., 2020; Kidambi et al., 2020; Swazinna et al., 2021b; Yu et al., 2021; Rigter et al., 2022; Swazinna et al., 2021a). Even approaches that train policies purely supervised, by conditioning on performance, have been proposed (Peng et al., 2019; Emmons et al., 2021; Chen et al., 2021). Model based algorithms more often use model uncertainty, while model-free methods use a more direct behavior regularization approach.

**Offline policy evaluation** or offline hyperparameter selection is concerned with evaluating (or at least ranking) policies that have been found by an offline RL algorithm, in order to either pick the best performing one or to tune hyperparameters. Often, dynamics models are used to evaluate policies found in model-free algorithms, however also model-free evaluation methods exist (Hans et al., 2011; Paine et al., 2020; Konyushova et al., 2021; Zhang et al., 2021b; Fu et al., 2021). Unfortunately, but also intuitively, this problem is rather hard since if any method is found that can more accurately assess the policy performance than the mechanism in the offline algorithm used for training, it should be used instead of the previously employed method for training. Also, the general dilemma of not knowing in which parts of the state-action space we know enough to optimize behavior seems to always remain. Works such as Zhang et al. (2021a); Lu et al. (2021) become applicable if limited online evaluations are allowed, making hyperparameter tuning much more viable.

**Offline RL with online adaptation** Other works propose an online learning phase that follows after offline learning has conceded. In the most basic form, Kurenkov & Kolesnikov (2021) introduce an online evaluation budget that lets them find the best set of hyperparameters for an offline RL algorithm given limited online evaluation resources. In an effort to minimize such a budget, Yang et al. (2021) train a set of policies spanning a diverse set of uncertainty-performance trade-offs. Ma et al. (2021) propose a conservative adaptive penalty, that penalizes unknown behavior more during the beginning and less during the end of training, leading to safer policies during training. In Pong et al.

(2021); Nair et al. (2020); Zhao et al. (2021) methods for effective online learning phases that follow the offline learning phase are proposed. In contrast to these methods, we are not aiming for a fully automated solution. Instead, we want to provide the user with a valuable tool after training, so we do not propose an actual online phase, also since practitioners may find any performance deterioration inacceptable. To the best of our knowledge, no prior offline RL method produces policies that remain adaptable after deployment without any further training.

## 3 LION: Learning in Interactive Offline eNvironments

In this work, we address two dilemmas of the offline RL setting: First and foremost, we would like to provide the user with a high level control option in order to influence the behavior of the policy, since we argue that the user is crucially important for solving the task and not to be made obsolete by an algorithm. Further we address the issue that in offline RL, the correct hyperparameter controlling the trade-off between conservatism and performance is unknown and can hardly be tuned. By training a policy conditioned in the proximity hyperparameter, we aim to enable the user to find a good trade-off hyperparameter. Code will be made available at https://github.com/pswazinna/LION.

As mentioned, behavior cloning, will most likely yield the most trustworthy solution due to its familiarity, however the solution is of very limited use since it does not outperform the previous one. Offline RL on the other hand is problematic since we cannot simply evaluate policy candidates on the real system and offline policy evaluation is still an open problem (Hans et al., 2011; Paine et al., 2020; Konyushova et al., 2021; Zhang et al., 2021b; Fu et al., 2021). In the following, we thus propose a solution that moves the hyperparameter choice from training to deployment time, enabling the user to interactively find the desired trade-off between BC and offline optimization. A user may then slowly move from conservative towards better solutions.

### 3.1 Training

During training time, we optimize three components: A model of the original policy $\beta_\phi(s)$, an ensemble of transition dynamics models $\{f^i_{\psi_i}(s, a) | i \in 0, \ldots, N-1\}$, as well as the user adaptive policy $\pi_\theta(s, \lambda)$. The dynamics models $\{f^i\}$ as well as the original policy $\beta$ are trained in isolation before the actual policy training starts. Both $\pi$ and $\beta$ are always simple feedforward neural networks which map states directly to actions in a deterministic fashion (practitioners likely favor deterministic policies over stochastic ones due to trust issues). $\beta$ is trained to simply imitate the behavior present in the dataset by minimizing the mean squared distance to the observed actions:

$$L(\phi) = \frac{1}{N} \sum_{s_t, a_t \sim \mathcal{D}} [a_t - \beta_\phi(s_t)]^2 \tag{1}$$

Depending on the environment, the transition models are either also feedforward networks or simple recurrent networks with a single recurrent layer. The recurrent networks build their hidden state over $G$ steps and are then trained to predict a window of size $F$ into the future (similarly to (Hein et al., 2017b)), while the feedforward dynamics simply predict single step transitions. Both use mean squared error as loss:

$$L(\psi_i) = \frac{1}{N} \sum_{s_t, a_t, s_{t+1} \sim \mathcal{D}} \left[ s_{t+1} - f^i_{\psi_i}(s_t, a_t) \right]^2 \tag{2}$$

$$L(\psi_i) = \frac{1}{N} \sum_{t \sim \mathcal{D}} \sum_{f=1}^{F} [s_{t+G+f+1} - f^i_{\psi_i}(s_t, a_t, \ldots s_{t+G}, a_{t+G}, \ldots \hat{s}_{t+G+f}, a_{t+G+f})]^2$$

where $\hat{s}_{t+H+f}$ are the model predictions that are fed back to be used as input again. For simplicity, in this notation we assume the reward to be part of the state. Also we do not explicitly show the recurrence and carrying over of the hidden states.

After having trained the two components $\beta_\phi(s)$ and $\{f^i_{\psi_i}(s, a)\}$, we can then move on to policy training. Similarly to MOOSE and WSBC, we optimize the policy $\pi_\theta$ by sampling start states from $\mathcal{D}$ and performing virtual rollouts throughout the dynamics ensemble using the current policy candidate. In every step, the ensemble predicts the reward as the minimum among its members and the next state

that goes with it. At the same time we collect the mean squared differences between the actions that $\pi_\theta$ took in the rollout and the one that $\beta_\phi$ would have taken. The loss is then computed as a weighted sum of the two components. Crucially, we sample the weighting factor $\lambda$ randomly and pass it to the policy as an additional input - the policy thus needs to learn all behaviors ranging from pure behavior cloning to entirely free optimization:

$$L(\theta) = - \sum_{s_0 \sim \mathcal{D}} \sum_t^T \gamma^t [\lambda e(s_t, a_t) - (1 - \lambda)p(a_t)] \qquad a_t = \pi_\theta(s_t, \lambda) \qquad (3)$$

where we sample $\lambda$ between 0 & 1, $e(s_t, a_t) = \min\{r(f^i_{\psi^i}(s_t, a_t))|i \in 0, ..., N - 1\}$ denotes the output of the ensemble prediction for reward (we omit explicit notation of recurrence for simplicity) and $p(a_t) = [\beta_\psi(s_t) - a_t]^2$ denotes the penalty based on the mean squared distance between the original policy and the actions proposed by $\pi_\theta$. See Fig. 1 for a visualization of our proposed training procedure.

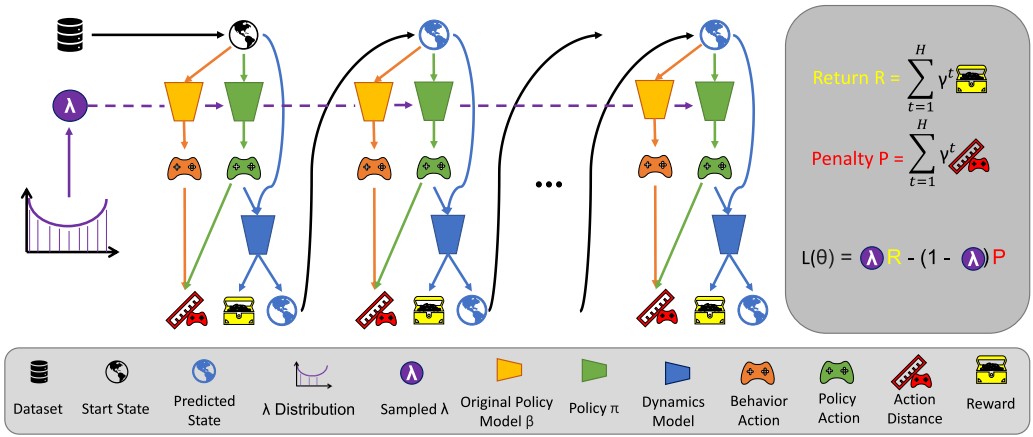

Figure 1: Schematic of LION policy training. During policy training (Eq. 3) only $\pi_\theta$ (in green) is adapted, while the original policy model $\beta_\phi$ (orange) and the dynamics ensemble $\{f_{\psi^i}\}$ (blue) are already trained and remain unchanged. From left to right, we first sample a start state (black) from the dataset and a $\lambda$ value from its distribution. Then, we let the original policy (orange) as well as the currently trained policy (green) predict actions - note that the newly trained policy is conditioned on $\lambda$. Both actions are then compared to calculate the penalty for that timestep (red). The action from the currently trained policy is then also fed into the trained transition model (blue) together with the current state (black / blue), to get the reward for that timestep (yellow) as well as the next state (blue). This procedure is repeated until the horizon of the episode is reached. The rewards and penalties are then summed up and weighted by $\lambda$ to be used as a loss function for policy training.

We motivate our purely model-based approach (no value function involved) with the fact that we have fewer moving parts: Our ensemble can be kept fixed once it is trained, while a value function has to be learned jointly with $\pi_\theta$, which is in our case more complex than usual. See experimental results in Fig. 10 a brief attempt at making our approach work in the model-free domain.

In addition to Eq. 3, we need to penalize divergence not only from the learned model of the original policy during virtual roll-outs, but also from the actual actions in the dataset at $\lambda = 0$. It seems that if this is not done, the trained policy $\pi$ sticks to the (also trained) original policy $\beta$ during the rollouts, but during those rollouts, there are states that did not appear in the original dataset, enabling $\pi$ to actually diverge from the true trajectory distribution. We thus penalize both rollout as well as data divergence at $\lambda = 0$:

$$L(\theta) = - \sum_{s_0 \sim \mathcal{D}} \sum_t^T \gamma^t [\lambda e(s_t, a_t) - (1 - \lambda)p(a_t)] \quad + \quad \eta \sum_{s,a \sim \mathcal{D}} [a - \pi(s, \lambda = 0)]^2 \qquad (4)$$

where $\eta$ controls the penalty weight for not following dataset actions at $\lambda = 0$, see Appendix A for more details. Furthermore, we normalize states to have zero mean and unit variance during every

forward pass through dynamics model or policy, using the mean and standard deviation observed in the dataset. We also normalize the rewards provided by the ensemble $r_t = e(s_t, a_t)$, so that they live in the same magnitude as the action penalties (we assume actions to be in $[-1, 1]^D$, so that the penalty can be in $[0, 4]^D$ where $D$ is the action dimensionality).

Intuitively, one might choose to sample $\lambda$ uniformly between zero and one, however instead we choose a beta distribution with parameters $(0.1, 0.1)$, which could be called bathtub-shaped. Similarly to (Seo et al., 2021), we find that it is important to put emphasis on the edge cases, so that the extreme behavior is properly learned, rather than putting equal probability mass on each value in the [0, 1] range. The interpolation between the edges seems to be easier and thus require less samples. Fig. 11 shows policy results for different lambda distributions during training.

## 3.2 Deployment

At inference time, the trained policy can at any point be influenced by the user that would otherwise be in control of the system, by choosing the $\lambda$ that is passed to the policy together with the current system state to obtain an action:

$$a_t = \pi_\theta(s_t, \lambda) \quad \lambda \in \text{User}(s_t). \quad (5)$$

He or she may choose to be conservative or adventurous, observe the feedback and always adjust the proximity parameter of the policy accordingly. At this point, any disliked behavior can immediately be corrected without any time loss due to re-training and deploying a new policy, even if the user's specific preferences were not known at training time.

We propose to initially start with $\lambda = 0$ during deployment, in order to check whether the policy is actually able to reproduce the original policy and to gain the user's trust in the found solution. Then, depending on

---

**Algorithm 1** LION (Training)

1: **Require** Dataset $D = \{\tau_i\}$, randomly initialized parameters $\theta, \phi, \psi$, lambda distribution parameters $\text{Beta}(a, b)$, horizon $H$, number of policy updates $U$
2: // dynamics and original policy models can be trained supervised and independently of other components
3: train original policy model $\beta_\phi$ using $D$ and Equation 1
4: train dynamics models $f_{\psi^i}^i$ with $D$ and Equation 2
5: **for** j in 1..U **do**
6:     sample start states $S_0 \sim D$
7:     sample lambda values $\lambda \sim \text{Beta}(a, b)$
8:     initialize policy loss $L(\theta) = 0$
9:     **for** t in 0..H **do**
10:         calculate policy actions $a_t = \pi_\theta(s_t, \lambda)$
11:         calculate behavioral actions $b_t = \beta_\phi(s_t)$
12:         calculate penalty term $p(a_t) = [\beta_\psi(s_t) - a_t]^2$
13:         $r_t, s_{t+1} = f_{\psi^i}^i(s_t, a_t)$
            $s.t. \quad i = \arg\min_i\{r(f_{\psi^i}^i(s_t, a_t))\}$
14:         $L(\theta)+ = -\gamma^t[\lambda r_t - (1 - \lambda)p(a_t)]$
15:     update $\pi_\theta$ using gradient $\nabla_\theta L(\theta)$ and Adam
16: **return** $\pi_\theta$;

---

how critical failures are and how much time is at hand, $\lambda$ may be increased in small steps for as long as the user is still comfortable with the observed behavior. Figure 3 shows an example of how the policy behavior changes over the course of $\lambda$. Once the performance stops to increase or the user is otherwise not satisfied, we can immediately return to the last satisfying $\lambda$ value.

## 4 Experiments

At first, we intuitively showcase LION in a simple 2D-world in order to get an understanding of how the policy changes its behavior based on $\lambda$. Afterwards, we move to a more serious test, evaluating our algorithm on the 16 industrial benchmark (IB) datasets (Hein et al., 2017a; Swazinna et al., 2021b). We aim to answer the following questions:

- Do LION policies behave as expected, i.e. do they reproduce the original policy at $\lambda = 0$ and deviate more and more from it with increased freedom to optimize for return?
- Do LION policies at least in parts of the spanned $\lambda$ space perform better or similarly well to state of the art offline RL algorithms?
- Is it easy to find the $\lambda$ values that maximize return for practitioners? That is, are the performance courses smooth or do they have multiple local mini- & maxima?
- Is it possible for users to exploit the $\lambda$ regularization at runtime to restrict the policy to only exhibit behavior he or she is comfortable with?

## 4.1 2D-WORLD

We evaluate the LION approach on a simplistic 2D benchmark. The states are x & y coordinates in the environment and rewards are given based on the position of the agent, following a Gaussian distribution around a fixed point in the state space, i.e. $r(s_t) = \frac{1}{\sigma\sqrt{2\pi}}e^{-0.5((s_t-\mu)/\sigma)^2}$. In this example we set $\mu = (3,6)^{\mathrm{T}}$ and $\sigma = (1.5, 1.5)^{\mathrm{T}}$. A visualization of the reward distribution can be seen in Fig. 2 (b). We collect data from the environment using a simple policy that moves either to position $(2.5, 2.5)^{\mathrm{T}}$ or to

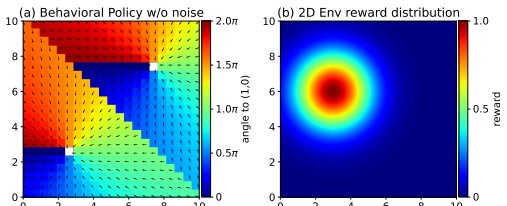

Figure 2: (a) Original policy for data collection and - color represents action direction (b) reward distribution in the 2D environment - color represents reward value

$(7.5, 7.5)^{\mathrm{T}}$, depending on which is closer to the randomly drawn start state (shown in Fig. 2(a)), adding $\varepsilon = 10\%$ random actions as exploration. Then we follow the outlined training procedure, by training a transition model, original policy model and finally a new policy that can at runtime change its behavior based on the desired proximity to the original policy. Fig. 3 shows policy maps for $\lambda \in \{0.0, 0.6, 0.65, 0.7, 0.85, 1.0\}$, moving from simply imitating the original policy, over different mixtures, to pure return optimization. Since the task is easy and accurately modeled by the dynamics ensemble, one may give absolute freedom to the policy and optimize for return only. As it can be seen, the policy moves quickly to the center of the reward distribution for $\lambda = 1$.

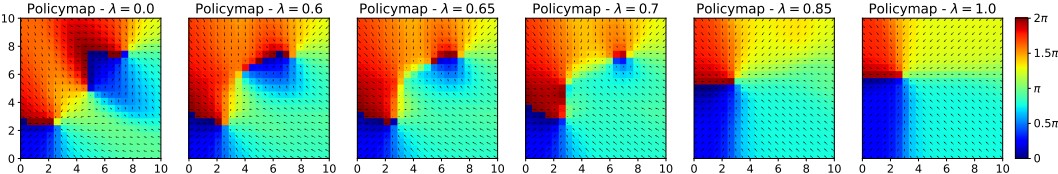

Figure 3: Policy maps for increasing values of $\lambda$ in the 2D environment - colors represent action direction. Initially, the policy simply imitates the original policy (see Fig. 2 (a)). With increased freedom, the policy moves less to the upper right and more to the bottom left goal state of the original policy, since that one is closer to the high rewards. Then, the policy moves its goal slowly upwards on the y-axis until it is approximately at the center of the reward distribution. Since enough data was available (1,000 interactions) and the environment so simple, the models capture the true dynamics well and the optimal solution is found at $\lambda = 1$. This is however not necessarily the case if not enough or not the right data was collected (e.g. due to a suboptimal original policy - see Fig. 4).

## 4.2 INDUSTRIAL BENCHMARK

**Datasets** We evaluate LION on the industrial benchmark datasets initially proposed in (Swazinna et al., 2021b). The 16 datasets are created with three different baseline original policies (*optimized*, *mediocre*, *bad*) mixed with varying degrees of exploration. The *optimized* baseline is an RL trained policy and simulates an expert practitioner. The *mediocre* baseline moves the system back and forth around a fixed point that is rather well behaved, while the *bad* baseline steers to a point on the edge of the state space in which rewards are deliberately bad. Each baseline is combined with $\varepsilon \in \{0.0, 0.2, 0.4, 0.6, 0.8, 1.0\}$-greedy exploration to collect a dataset (making the $\varepsilon = 0.0$ datasets extreme cases of the narrow distribution problem). Together, they constitute a diverse set of offline RL settings. The exact baseline policies are given by:

$$\pi_{\mathrm{bad}} = \begin{cases} 100 - v_t \\ 100 - g_t \\ 100 - h_t \end{cases} \quad \pi_{\mathrm{med}} = \begin{cases} 25 - v_t \\ 25 - g_t \\ 25 - h_t \end{cases} \quad \pi_{\mathrm{opt}} = \begin{cases} -\tilde{v}_{t-5} - 0.91 \\ 2\tilde{f}_{t-3} - \tilde{p} + 1.43 \\ -3.48\tilde{h}_{t-3} - \tilde{h}_{t-4} + 2\tilde{p} + 0.81 \end{cases}$$

The datasets contain 100,000 interactions collected by the respective baseline policy combined with the $\varepsilon$-greedy exploration. The IB is a high dimensional and partially observable environment - if access to the full Markov state were provided, it would contain 20 state variables. Since only six of those are observable, and the relationship to the other variables and their subdynamics are complex

and feature heavily delayed components, prior work Hein et al. (2017b) has stated that up to 30 past time steps are needed to form a state that can hope to recover the true dynamics, so the state can be considered 180 dimensional. In our case we thus set the number of history steps $G = 30$. The action space is 3 dimensional. The benchmark is not supposed to mimic a single industrial application, but rather exhibit common issues observable in many different applications (partial observability, delayed rewards, multimodal and heteroskedastic noise, ...). The reward is a weighted combination of the observable variables fatigue and consumption, which are conflicting (usually move in opposite directions and need trade-off) and are influenced by various unobservable variables. As in prior work Hein et al. (2018); Depeweg et al. (2016); Swazinna et al. (2021b) we optimize for a horizon of 100. The datasets are available at `https://github.com/siemens/industrialbenchmark/tree/offline_datasets/datasets` under the Apache License 2.0.

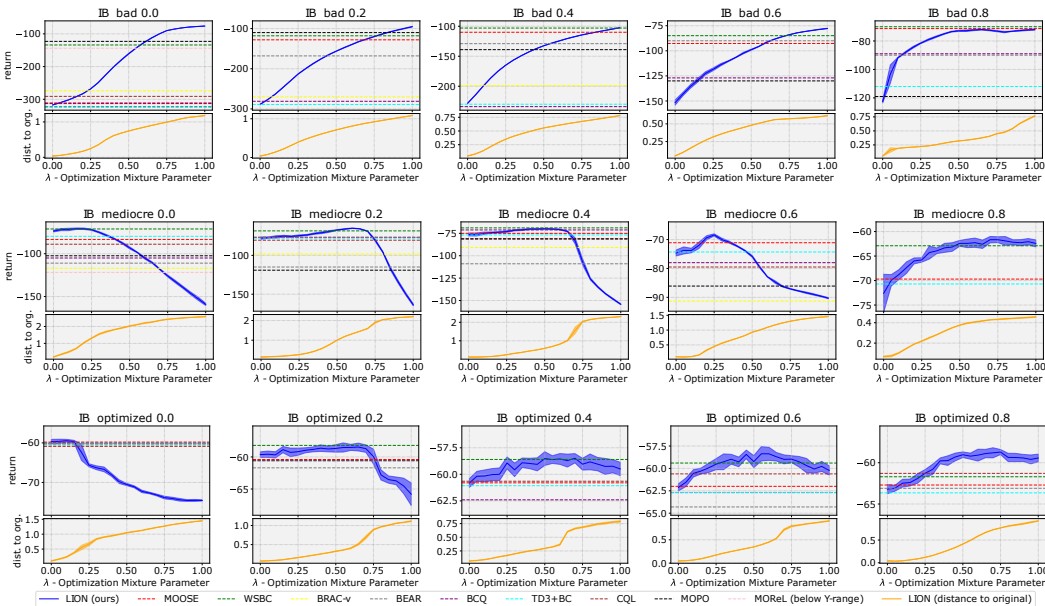

Figure 4: Evaluation performance (top portion of each graph) and distance to the original policy (lower portion of each graph) of the LION approach over the chosen $\lambda$ hyperparameter. Various state of the art baselines are added as dashed lines with their standard set of hyperparameters (results from (Swazinna et al., 2022)). Even though the baselines all exhibit some hyperparameter that controls the distance to the original policy, all are implemented differently and we can neither map them to a corresponding lambda value of our algorithm, nor change the behavior at runtime, which is why we display them as dashed lines over the entire $\lambda$-spectrum. See Fig. 12 for the 100% exploration dataset.

**Baselines** We compare performances of LION with various state of the art offline RL baselines:

- BEAR, BRAC, BCQ, CQL and TD3+BC (Kumar et al., 2019; Wu et al., 2019; Fujimoto et al., 2019; Kumar et al., 2020; Fujimoto & Gu, 2021) are model-free algorithms. They mostly regularize the policy by minimizing a divergence to the original policy. BCQ samples only likely actions and CQL searches for a Q-function that lower bounds the true one.

- MOOSE and WSBC (Swazinna et al., 2021a;b) are purely model based algorithms that optimize the policy via virtual trajectories through the learned transition model. MOOSE penalizes reconstruction loss of actions under the original policy (learned by an autoencoder), while WSBC constrains the policy directly in weight space. MOOSE is from the policy training perspective the closest to our LION approach.

- MOPO and MOReL (Yu et al., 2020; Kidambi et al., 2020) are hybrid methods that learn a transition model as well as a value function. Both use the models to collect additional data and regularize the policy by means of model uncertainty. MOPO penalizes uncertainty directly, while MOReL simply stops episodes in which future states become too unreliable. MOReL uses model-disagreement and MOPO Gaussian outputs to quantify uncertainty.

**Evaluation** In order to test whether the trained LION policies are able to provide state of the art performance anywhere in the $\lambda$ range, we evaluate them for $\lambda$ from 0 to 1 in many small steps. Figs. 4 and 12 show results for the 16 IB datasets. We find that the performance curves do not exhibit many local optima. Rather, there is usually a single maximum before which the performance is rising and after which the performance is strictly dropping. This is a very desirable characteristic for usage in the user interactive setting, as it enables users to easily find the best performing $\lambda$ value for the policy. In 13 out of 16 datasets, users can thus match or outperform the current state of the art method on that dataset, and achieve close to on-par performance on the remaining three. The distance-to-original-policy curves are even monotonously increasing from start to finish, making it possible for the practitioner to find the best solution he or she is still comfortable with in terms of distance to the familiar behavior.

**Discrete baseline** A simpler approach might be to train an existing offline RL algorithm for many trade-offs in advance, to provide at least discrete options. Two downsides are obvious: (a) we wouldn't be able to handle the continuous case, i.e. when a user wants a trade-off that lies between two discrete policies, and (b) the computational cost increases linearly with the number of policies trained. We show that a potentially even bigger issue exists in Figure 5: When we train a discrete collection of policies with different hyperparameters, completely independently of each other, they often exhibit wildly different behaviors even when the change in hyperparameter was small. LION instead expresses the collection as a single policy network, training them jointly and thus forcing them to smoothly interpolate among each other. This helps to make the performance a smooth function of the hyperparameter (although this must not always be the case) and results in a performance landscape that is much easier to navigate for a user searching for a good trade-off.

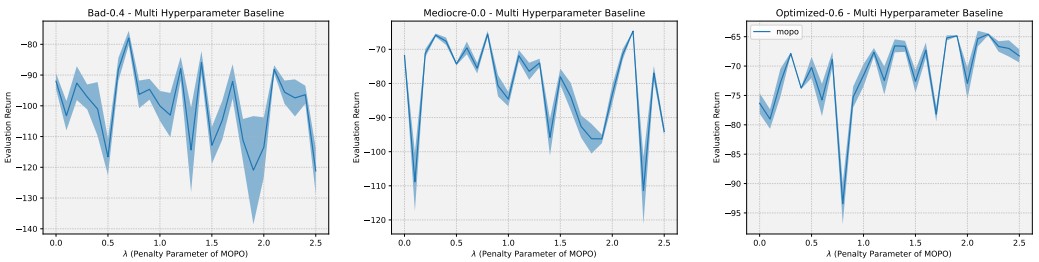

Figure 5: Prior offline RL algorithms like MOPO do not behave consistently when trained across a range of penalizing hyperparameters.

**Return conditioning baseline** Another interesting line of work trains policies conditional on the return to go, such as RvS (Emmons et al., 2021) (Reinforcement Learning via Supervised Learning) or DT (Chen et al., 2021) (Decision Transformer). A key advantage of these methods is their simplicity - they require neither transition model nor value function, just a policy suffices, and the learning can be performed in an entirely supervised fashion. The resulting policies could be interpreted in a similar way as LION policies: Conditioning on returns close to the original performance would result in the original behavior, while choosing to condition on higher returns may lead to improved performance if the extrapolation works well. In Fig. 6 we report results of the RvS algorithm on the same datasets as the discrete baseline. The returns in the datasets do not exhibit a lot of variance, so it is unsurprising that the approach did not succeed in learning a lot of different behaviors.

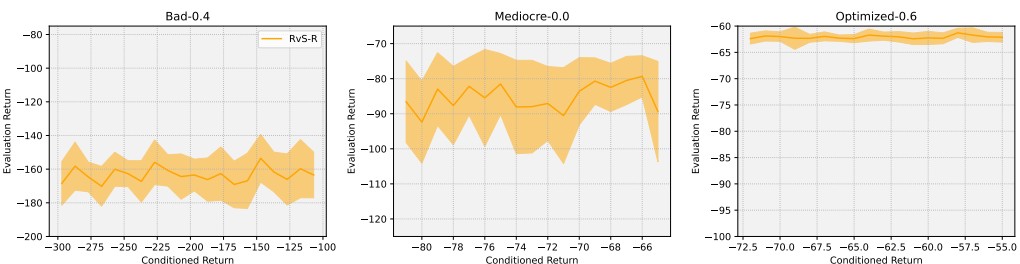

Figure 6: Return conditioned policies did not learn many different behaviors on the IB datasets.

**Finding a suitable $\lambda$** We would like to emphasize that we neither want to optimize **all** offline hyperparameters with our solution, nor are we interested in a fully automated solution. Users may thus adopt arbitrary strategies to finding their personal trade-off of preference. We will however provide a conservative example strategy: The operator starts with the most conservative value available and then moves in small, but constant steps towards more freedom. Whenever the performance drops below the previous best or the baseline performance, he immediately stops and uses the last $\lambda$ before that. Table 1 summarizes how this strategy would perform.

| Dataset | LION Final $\lambda$ | MOPO Final $\lambda$ | RvS Final $\hat{R}$ | LION Return | MOPO Return | RvS Return |
|---|---|---|---|---|---|---|
| Bad-0.4 | 1.0 | 2.4 | -287 | **-102.4** | -107.2 | -158.2 |
| Mediocre-0.0 | 0.2 | 2.4 | -81 | **-70.8** | -87.8 | -86.6 |
| Optimized-0.6 | 0.35 | 2.5 | -71 | **-58.9** | -72.4 | -61.9 |

Table 1: If a user adopts the simple strategy of moving in small steps (0.05 for LION, 0.1 for MOPO since its range is larger, 10.0 / 1.0 for RvS) from conservative towards better solutions, immediately stopping when a performance drop is observed, LION finds much better solutions due to the consistent interpolation between trade-offs. Note that in MOPO, we start with large $\lambda = 2.5$ ($1.0$ is the default) since there it controls the penalty, while we start with $\lambda = 0$ in LION, where it controls the return.

## 5    DISCUSSION & CONCLUSION

In this work we presented a novel offline RL approach that, to the best of our knowledge, is the first to let the user adapt the policy behavior after training is finished. We let the user tune the behavior by allowing him to choose the desired proximity to the original policy, in an attempt to solve two issues: (1) The problem that practitioners cannot tune the hyperparameter in offline RL & (2) the general issue that users have no high level control option when using RL policies (they might even have individual preferences with regards to the behavior of a policy that go beyond just performance).

We find that effectively, LION provides a high level control option to the user, while still profiting from a high level of automation. It furthermore takes much of the risk that users normally assume in offline RL away since deployments can always start with a BC policy when they start at $\lambda = 0$, before moving to better options. While behavior cloning does not have to work in general, we did not experience any issues with it in our experiments, and it should be easier than performing RL since it can be done entirely in a supervised fashion. Given that BC works, deployments can thus start with minimal risk. In prior offline algorithms, users experienced the risk that the algorithm did not produce a satisfying policy on the particular dataset they chose. E.g.: WSBC produces state of the art results for many of the IB datasets, however for mediocre-0.6 it produces a catastrophic -243 (original performance is -75). Similarly, CQL is the prior best method on optimized-0.8, however the same method produces a performance of -292 on bad-0.2 (MOOSE, MOPO, and WSBC get between -110 & -130). Due to the smoothness of the interpolation of behaviors in LION, practitioners should be able to use it to find better trade-offs with lower risk than prior methods. Adaptable policies are thus likely a step towards more deployments in industrial applications.

**Future Work** As outlined at the end of Section C of the appendix, we were unable to incorporate value functions into our approach. This can be seen as a limiting factor, since there exist environments with sparse or very delayed rewards or that for other reasons exhibit long planning horizons. The industrial benchmark features delayed rewards and evaluation trajectories are 100 steps long, however other environments can be more extreme in their characteristics. At some point, even the best dynamics models suffer from compounding errors and cannot accurately predict the far away future. We do not believe that it is in principle not possible to combine the LION approach with value functions, however future work will likely need to find methods to stabilize the learning process.

Other potential limitations of our approach include difficulties with the behavior cloning, e.g. when the original policy is stochastic or was not defined in the same state space as we use (e.g. different human operators controlled the system at different times in the dataset), as well as difficulties when interpolating between vastly different behaviors on the Pareto front spanned by proximity and performance. We mention these potential limitations only for the sake of completeness since we were unable to observe them in our practical experiments.

## 6 ETHICS STATEMENT

Generally, (offline) reinforcement learning methods can be used to optimize control policies for many different systems beyond the control of the researchers. Malicious individuals may thus use findings in manners that negatively impact others or society as a whole (e.g. by using it in weapons). Faulty policies could also potentially result in other safety concerns (e.g. policies for heavy machinery). In ML research, there is also always the concern that increased automation may lead to rising unemployment. While some institutions like the UN and the WEF are convinced AI will lead to long term job growth, Bordot (2022) found that a slightly increased unemployment risk can be found empirically.

Beyond these usual concerns that mostly apply to ML/RL research as a whole, rather than our individual paper, we believe that no additional considerations are necessary for our method. If at all, the societal impact should be positive, since our solution fosters trust by practitioners and should thus increase the usage and human-machine collaborative deployment of offline RL policies in practice, enabling more efficient control strategies in many industrial domains.

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

APPENDIX

## A  OTHER HYPERPARAMETERS

Similarly to other methods, we introduce a set of hyperparameters in our method either explicitly or implicitly. Any hyperparameter that is involved in training the original policy model $\beta_\phi$ or the transition dynamics models $\{f_{\psi^i}\}$ can obviously be tuned offline, by holding out a validation set of transitions from the original dataset and evaluating the trained models on them to test how well they predict dataset actions or future states and rewards. This is true for example for the architecture of the transition models, their learning rate, or the number of epochs to train.

When we train the policy in offline reinforcement learning however, the equivalent of an evaluation dataset does not exist, since we cannot evaluate the policy on the true environment dynamics. While efforts are being made to make hyperparameter tuning possible via offline policy evaluation, the problem is hard for obvious reasons and we cannot really rely on them yet. Especially the arguably most important hyperparameter, how closely the newly trained policy needs to follow the actions proposed by the original policy, cannot realistically be tuned since it involves knowing when the evaluation method itself cannot be trusted any more. This is precisely why we propose the selection of this hyperparameter at runtime via user interaction.

Our method however introduces some hyperparameters that are used during policy training and that could thus be seen as critical hyperparameters that cannot be tuned: $\eta$, controlling the penalty magnitude for not following the dataset actions at $\lambda = 0$, and the parameters of the Beta distribution from which $\lambda$ is sampled during training. We did not tune them, since we simply copied the parameters from (Seo et al., 2021) so that $\lambda \sim \text{Beta}(0.1, 0.1)$ and also set $\eta = 0.1$ without any experiments. We would however like to point out, that these parameters are at least partially tunable even in the offline setting, since they do not depend on the accuracy of the transition models: $\eta$ controls how strongly one adheres to the actions of the dataset, and we can easily asses whether this is working properly, by evaluating whether the final trained policy still adheres to the actions of the dataset at $\lambda = 0$. We evaluated policies for different $\eta$ in Figure 7 (a) and show that there is not a great difference between different $\eta$, as long as it is not zero, so it seems reasonable to choose something small that is still doing the trick. Similarly, we show an evaluation for different parameters of the beta distribution in Fig. 7 (b). The parameters control how often which parts of the $\lambda$-spectrum are seen during training (see Fig. 11). As can be seen, the more the distribution tends towards a uniform distribution, the more the policy deviates from the original policy model at $\lambda = 0$, which is undesired. When more probability mass is focused on the edges of the distribution, we can see that the original policy model is reproduced more accurately.

Both $\eta$ and $\beta$ can thus be reasonably well tuned entirely offline, even without using the transition models.

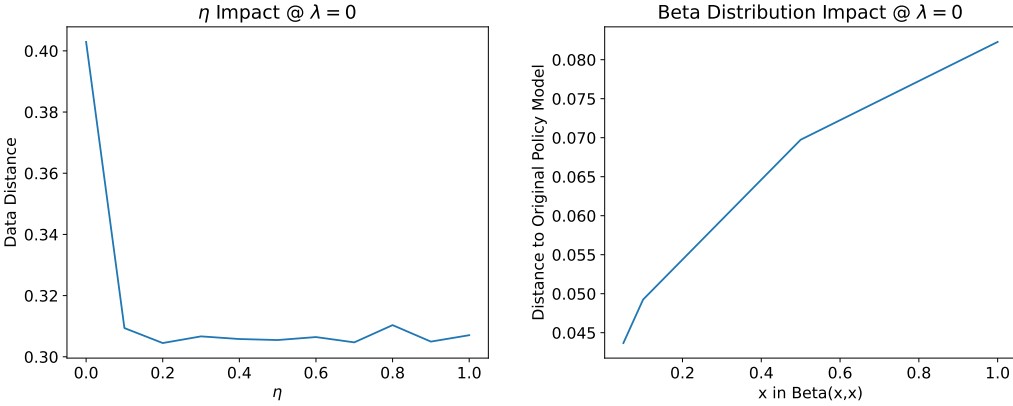

Figure 7: Some hyperparameters like $\eta$ and the parameters of the Beta distribution can be adjusted entirely offline, since they do not require evaluation on the true transition dynamics.

## B    COMPLETE POLICY VISUALIZATION IN 2D-ENV

In Fig. 3, we chose only a set of interesting $\lambda$ values for visualization for page limit reasons. See Fig 8 for a more fine-grained evaluation.

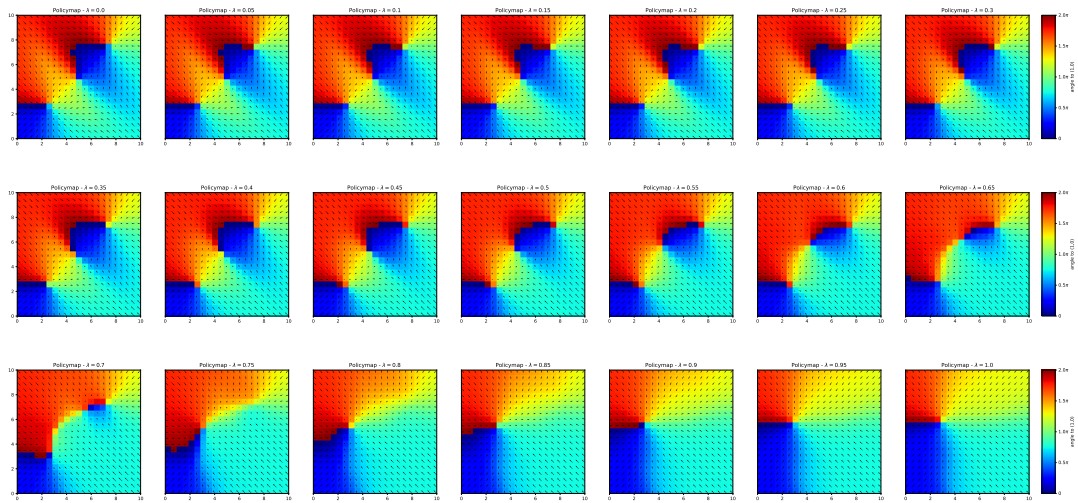

Figure 8: Visualization for more $\lambda$ values of the trained policy in the simple 2D Env than in Fig. 3.

## C    MODEL-FREE EXPERIMENTS

**Model-free** We motivate our purely model-based implementation with the idea that it should be simpler to implement compared to model-free policy training. When learning a policy with a value function, we have always two models that are trained jointly - the policy and the Q-function interact with each other and both are essentially a moving target. While it should be possible to derive a similar algorithm as LION in the model-free world, we were not quite able to do so: Figs. 9 and 10 show experiments in which we augment TD3+BC with our approach by sampling $\lambda$ from the same distribution as in LION and providing it to the policy, only that now it controls the influence the value function has on the optimization of the policy and $(1-\lambda)$ controls the distance to the behavior actions. Interestingly however, the trained policy is unable to alter its behavior anywhere in the $\lambda$-range. Instead it seems that the policy training is rather unstable and the policy collapses to a solution that behaves always the same, regardless of $\lambda$. Similarly, the distance to the original policy does not change with varying $\lambda$. We chose TD3+BC due to its simplicity - it exhibits fewer trained models and implementational pitfalls than others. The original Q-function and policy optimization objectives in TD3+BC are:

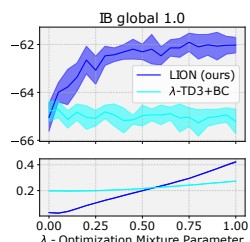

Figure 9: Model-free experiment on the 100% exploration dataset.

$$Q = \underset{Q}{\arg\min}\, \mathbb{E}_{s,a,s'\sim\mathcal{D}}\left[[Q(s,a) - (r + \gamma Q^t(s',\pi(s)))^2)]\right],$$

where $Q^t$ denotes the slow moving target Q-function used for computing the Q-targets. And:

$$\pi = \underset{\pi}{\arg\max}\, \mathbb{E}_{s,a\sim\mathcal{D}}\left[\frac{\alpha}{\frac{1}{N}\sum_s |Q(s,\pi(s))|}Q(s,\pi(s)) - (\pi(s) - a)^2\right]$$

In our extension, denoted by $\lambda$-TD3+BC, we provide the sampled $\lambda$ to the policy and use it to balance the two optimization terms, similarly to our LION approach. The new objectives are:

$$Q = \underset{Q}{\arg\min}\, \mathbb{E}_{s,a,s'\sim\mathcal{D}}\left[[Q(s,a,\boxed{\lambda}) - (r + \gamma Q^t(s',\pi(s,\boxed{\lambda}),\boxed{\lambda}))^2)]\right],$$

and:

$$\pi = \arg\max_{\pi} \mathbb{E}_{s,a\sim\mathcal{D}} \left[ \boxed{\lambda} \frac{\alpha}{\frac{1}{N}\sum_{s,a}|Q(s,\pi(s,\boxed{\lambda}),\boxed{\lambda})|} Q(s,\pi(s,\boxed{\lambda}),\boxed{\lambda}) - \boxed{(1-\lambda)}(\pi(s,\boxed{\lambda})-a)^2 \right]$$

We show complete experiments for all of the IB datasets in Figs. 10 & 9. As already mentioned in Section C, we are unable to observe that the policy is able to alter its behavior anywhere in the $\lambda$-range. We hypothesize that methods for stabilizing the Q-function are needed in order for the policy to not collapse.

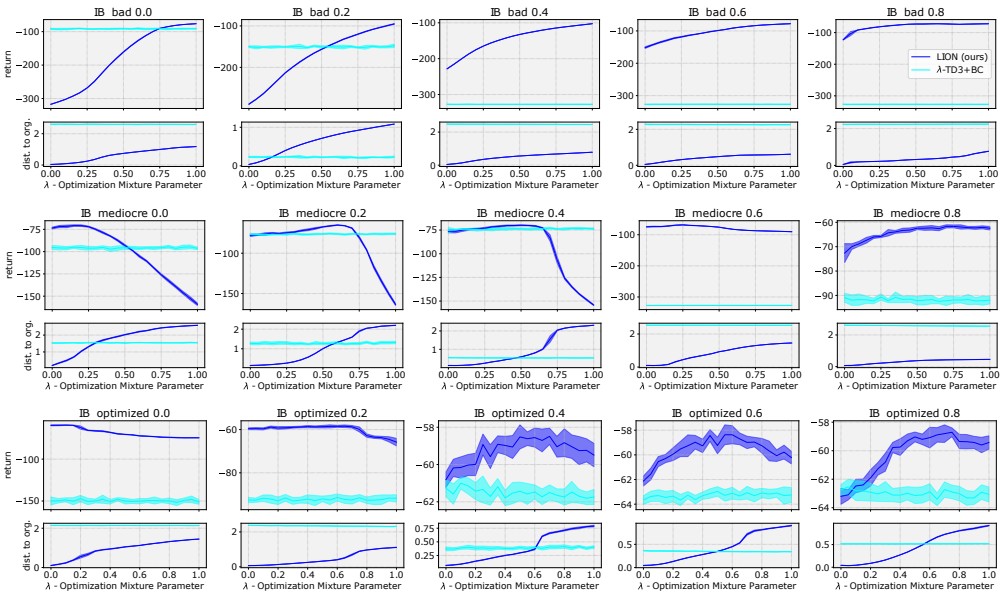

Figure 10: Model-free experiments incorporating the LION approach. The trained policies are unable to alter their behavior for different inputs of $\lambda$.

## D   SAMPLING $\lambda$

As mentioned, we do not sample $\lambda$ uniformly since doing so leads to less accurate learning at the edges of the $\lambda$-spectrum. In Fig 11, we show visualizations of different symmetric Beta distributions from which we draw $\lambda$ together with the policy results on the bad-0.2 dataset. The Beta(1, 1) case corresponds to the uniform distribution and has a significant discrepancy at the $\lambda = 0$ end of the range: The original policy (its performance) is not accurately reproduced. Generally, it seems the flatter the distribution becomes, the more are the two extreme cases moved together. We observe this phenomenon even though the policy has plenty of capacity and even in the simple 2D environment.

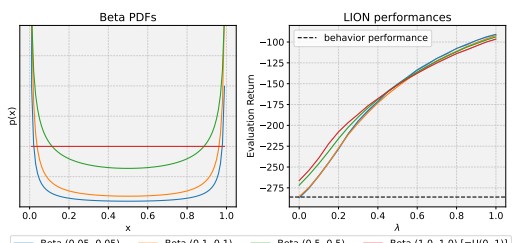

Figure 11: Beta distributions and resulting performances on bad-0.2. Edge cases become more accurate with increased probability mass.

# E    EXPERIMENTAL DETAILS

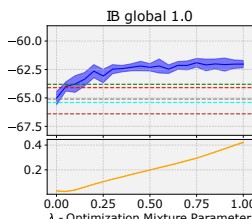

Figure 12: LION results as presented in Fig. 4 for the 100% exploration dataset.

We conducted experiments on a system with a Xeon Gold 5122 CPU ($4 \times 3.6$ GHz, no GPU support used). A single policy training took about 45 minutes, a single dynamics model about 2 hours. We train dynamics models using a 90/10 random data split and select the best models according to their validation performance. The final ensemble contains four models. The recurrent models for the industrial benchmark have an RNN cell with size 30 and an output layer mapping from the cell state to the state space. We use $G = 30$ history steps to build up the hidden state of the RNN and then predict $F = 50$ steps into the future. The feedforward models of the 2D env have two layers of size 20 & 10. We use adam (Kingma & Ba, 2014) with standard parameters and an exponentially decaying learning rate with factor $0.99$ and train for up to 3000 epochs. The models of the original policy have a single hidden layer of size 30 and the final policies have two layers of size 1024. We use ReLU nonlinearities throughout all experiments. During policy and original policy model training we use the entire dataset instead of just the 90% split from the dynamics training. We use a discount factor of $\gamma = 0.97$ and perform rollouts of length $H = 100$ to train and evaluate the policy.

# F    HYPERPARAMETERS OF THE TRANSITION ENSEMBLE

Instead of using the minimum over the predicted rewards during rollouts, one might as well use other forms of aggregation, like the mean. In Fig. 13 we show results for such mean reward experiments - the future state was then randomly selected among the predictions of the transition ensemble. The resulting performance curves look similar as those in 4, except that in the mediocre and the optimized setting, the performance decrease on the $\lambda = 1$ end is even steeper than with the minimum reward prediction. This makes intuitively sense, since the mean is much easier exploited than the minimum.

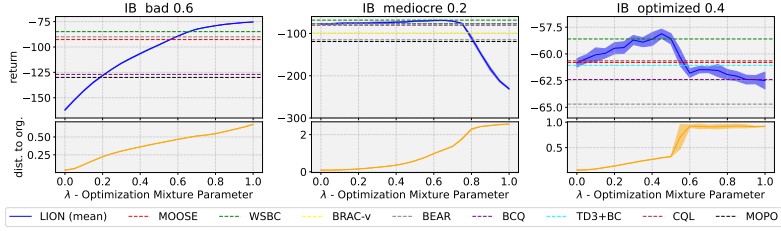

Figure 13: Results as in Fig. 4 for the mean reward rollout experiments.

Similar effects can be observed when the number of ensemble members is reduced. In Fig. 14, we show results on the same datasets when only a single model instead of an ensemble with four members is used (thus mean and min become the same). Here, the performance drops even earlier as well as stronger in the mediocre and optimized settings.

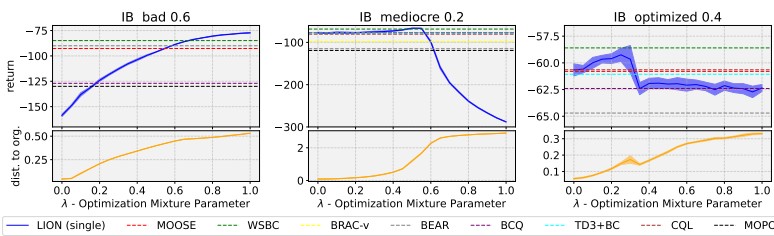

Figure 14: Results as in Fig. 4 for the single model rollout experiments.

## G    Preliminaries on Offline RL

In **offline reinforcement learning**, we are trying to optimize the sum of discounted rewards, called return, $R = \sum_t \gamma^t r_t$ obtained by acting in the MDP $\mathcal{M} = <S, A, R, T, \gamma, s_0>$, where $S$ is the set of states, $s_0 \subseteq S$ the set of starting states, $A$ the set of actions, $R : S \times A \to \mathbb{R}$ the reward function, $T : S \times A \to S$ the transition dynamics function, and $\gamma \in [0, 1)$ the discount factor. We train a policy $\pi_\theta : S \to A$ to obtain an action $a$ when in state $s$, in order to produce trajectories of length $H$: $\tau = (s_0, a_0, r_0, s_1, a_1, r_1, \dots, s_H, a_H, r_H)$. While online RL methods are able to continuously explore their environment, collect new data and test new behaviors, offline RL methods need to optimize the return $R$ by learning only from a fixed dataset of interactions $\mathcal{D} = \{\tau_i\}$, $i = 1, \dots, K$. Often, this is done by solving the approximate MDP $\hat{\mathcal{M}} = <S, A, R, T_\mathcal{D}, \gamma, s_0>$, where $T_\mathcal{D}$ is directly estimated with a transition model using the transitions from $\mathcal{D}$. However, also model-free offline approaches exist, that do not estimate $T_\mathcal{D}$ and directly try to optimize a policy with a value function that is only trained on $\mathcal{D}$. Since neither model-free nor model-based variants can be sure to accurately assess the policy's performance in unknown parts of the state-action space, offline RL algorithms regularize their policy to stay in a way 'close' to the policy that generated the data. The crucial question - **how close** should $\pi_\theta$ be to the original policy? - is an open problem. We address the dilemma that the user has no meaningful way of choosing the correct proximity since offline policy selection / evaluation remain generally unsolved problems.

