# OpenReview forum: "User-Interactive Offline Reinforcement Learning"
_ICLR.cc/2023/Conference — ICLR 2023 poster_

### Official Review · Reviewer_Kz6Z · 2022-10-23

**Confidence:** 4
**Correctness:** 4
**Technical Novelty And Significance:** 4
**Empirical Novelty And Significance:** Not applicable
**Recommendation:** 8

**Clarity, Quality, Novelty And Reproducibility:**

**Clarity:** This paper is very clearly written with beautiful figures; nicely done!

**Quality:** Sec 3 clearly lays out the main approach, with clear mathematical details, algorithmic descriptions, and figures for illustration. The experiments are also clearly documented.

**Novelty:** As far as this reviewer is aware, the proposed approach that learns an user-adaptive policy - though arguably quite simple but requires a change in perspective - has not been used in existing literature.

**Reproducibility:** Reference to existing datasets is provided. Algorithm description is also clear and should aid in reproducing the experiments. It would be great if the authors can share example implementation of their proposed algorithm.

**Strength And Weaknesses:**

**Strengths:**
- Problem is well motivated and well situated in the existing literature.
- The proposed approach requires some change in perspective (on how the trade-off parameter is used) but is intuitive to understand and simple to implement.
- Experiments on simulated problems and benchmarks are thoroughly executed with strong empirical results that support the main claims.

**Weaknesses:**
- Experiments are conducted on a industrial benchmark suite, could you further comment on the characteristics of the MDPs considered? e.g. deterministic vs stochastic transition, discrete or continuous states/actions? How much do these benchmark-specific characteristics affect the extent we can generalize the conclusions to other problems, for example medical domains and education (intelligent tutoring systems).

**minor comment on a parenthetical note:** on page 3 the authors state "practitioners likely favor deterministic policies over stochastic ones due to trust issues". Some past work have discussed the benefit of using non-deterministic (not the same as stochastic) policies and how they can be incorporated into a human-in-the-loop decision support system, which can be seen as another form of online user-interactive adaptation. It would be interesting to include a discussion in related works.
> - Fard & Pineau. "Non-deterministic policies in MDPs". JAIR 2011. https://doi.org/10.1613/jair.3175
> - Tang et al. "Clinician-in-the-Loop Decision Making: RL with Near-Optimal Set-Valued Policies". ICML 2020. https://proceedings.mlr.press/v119/tang20c.html


**Summary Of The Paper:**

This paper proposes a new offline RL learning procedure (named LION) that, at inference time, is responsive to a trade-off hyperparameter that controls how much the new policy can deviate from observed behavior; past work typically treats this as a hyperparameter at training time.

Experiments all in the offline setting, are conducted on a goal-finding task in 2D grid-world and an industrial control benchmark. Results show that the proposed approach can smoothly interpolate between imitating observed behavior and purely optimizing rewards, and that simple tuning strategies can help users find the optimal trade-off hyperparameter and also outperform existing offline RL methods.

**Summary Of The Review:**

This is a well written paper with simple yet novel proposed method, strong empirical results, and clear writing.

---

### Official Review · Reviewer_f6wy · 2022-10-25

**Confidence:** 4
**Correctness:** 4
**Technical Novelty And Significance:** 2
**Empirical Novelty And Significance:** 2
**Recommendation:** 3

**Clarity, Quality, Novelty And Reproducibility:**

- Some aspects about how the model is finally used are unclear
- the use of deterministic policies and world models is a clear restrictions
- Experiments are notmade on classical benchmarks
- The novelty is not big.

**Strength And Weaknesses:**

First of all, I must say that I don't really capture the path from the idea to the execution of the idea. If the objective is to learn a continuum of policies between pure imitation and reward maximization ones, I would suggest the authors to study for instance a loss function that would weight the behavioral cloning loss together with a reward driven loss (like IQL for instance, or CQL). Right now, the authors are doing this through a simulator of the world (deterministic in their case) which is far from being able to model complex problems. If the weighted loss is defined over the dataset, then the method becomes very close to the TD3+BC approach, and the originality would be to take the hyper parameter as an input to define the continuum of policies.

The second critical point is about what we can do with the resulting model. So, as an output, we have a policy that we can control with lambda. How do we concretely control the value of lambda at inference time? Maybe, for some values of lambda, the resulting policy is very good, but we don't know it. It seems to me that the value of lambda will then be chosen by using some kind of A/B testing (am I right ?). But in that case, it would mean that we have collected more data, and we could thususe these new data to update our policy...  So I don't understand exactly how the model is concretely used at last.

Last point, the experiments are made on not classical benchmarks. It is critical to achieve experiments on D4RL for instance to clearly compare with state of the art, particularly the fact that the authors are learning a world model in a very naive way which may have strong consequences on D4RL.

**Summary Of The Paper:**

The article describes a training strategy where one wants to obtain (from a dataset of episodes) a policy that is controllable by the user. The underlying assumption is to have the user in the loop at inference time (for some industrial reasons) instead of just outputting one single policy. Said otherwise, the authors want to put more weight on the human when the policy is deployed. To achieve this, they propose to proceed in two phases: in the first phase, they learn a behavioral cloning policy over the dataset, but also a world model to extract a simulator of the world. In a second step, they train a policy to maximize the reward on simulated trajectories but also add a regularization to enforce the new policy to potentially behave like the previously learned imitation policy (kind of distillation). What is interesting is that the weight of the regularization is considered as an output of the trained policy such that, as an output, one will obtain a continuum of policies, from pure (distilled) imitation policy to reward maximization ones. Experiments are made on a toy 2D problem, but also on some industrial benchmarks.

**Summary Of The Review:**

On my side, I don't really catch the interest of the proposed method and don't see it as a strong enough contribution.

---

### Official Review · Reviewer_edim · 2022-10-29

**Confidence:** 3
**Correctness:** 4
**Technical Novelty And Significance:** 3
**Empirical Novelty And Significance:** 3
**Recommendation:** 6

**Clarity, Quality, Novelty And Reproducibility:**

Clarity - The paper is clearly written, and the problem statement is well-motivated. The proposed method is clear.
Quality - The experiments are thorough.
Novelty - To my knowledge, the specific problem statement is novel, and relevant. The proposed method is straightforward, learning a policy that conditions on an additional input that controls proximity to the behavior policy (novelty here somewhat low).
Reproducibility - Experimental details are provided, however code is not, somewhat limiting the reproducibility of this work.

Nit
Figure 1 is visually hard to parse, and the caption is extremely long.
In Figures 2 and 3 it is very difficult to see the arrows. In Figure 3 the axes labels and titles are far too small.

**Strength And Weaknesses:**

Questions and Comments:
1) Why do you use the min of the ensemble instead of the expectation? This seems like a form of pessimism to avoid model exploitation, but if the environment is stochastic, the resulting policy would be biased. How important is this decision?
2) What is the effect of the ensemble experimentally? Is this necessary for the method to work?
3) In the middle row of Figure 4 (“mediocre” setting), why does performance decrease dramatically with increasing \lambda? Is this a case of the learned model being exploited? I’m surprised this doesn’t happen as dramatically in the bottom row (“optimized” setting).



**Summary Of The Paper:**

Summary: This paper proposes the problem of user-interactive offline RL. Specifically, a setting where the user can control the proximity of the learned policy to the behavior policy during deployment of the learned policy. The paper proposes a simple algorithm that provides this control, and compares it to standard offline RL and behavior cloning algorithms on a benchmark of industrial datasets.


Contributions:
 - Proposes the problem statement of offline RL that can be interactively tuned by a human user during deployment. While human-in-the-loop RL, online fine-tuning of offline RL policies, and meta-offline RL involve adapting the policy during deployment, I believe the specific aim to allow a human to control the interpolation between the behavior policy and a reward-optimized policy is novel.
Proposes a straightforward algorithm that builds on top of existing model-based offline RL algorithms.
Performs a thorough experimental evaluation on the industrial datasets benchmarks, comparing to a number of offline RL and BC algorithms.


**Summary Of The Review:**

Overall, the problem statement is well-motivated and practical. I think there are many applications in which we would like to do better than the expert, but have varying tolerance for doing worse (per state and per user) - this is a practical algorithms that makes it easy to tune this trade-off during deployment, without re-training. The method is a straightfoward combination of existing components, and experimental results show it is effective (caveat: I'm not familiar with the industrial benchmark used for experiments).



----- Update 12/9 ------

I have read the author response and the other reviews, and I maintain my positive assessment of this paper.

---

### Official Review · Reviewer_u1gw · 2022-11-01

**Confidence:** 4
**Correctness:** 4
**Technical Novelty And Significance:** 3
**Empirical Novelty And Significance:** 4
**Recommendation:** 10

**Clarity, Quality, Novelty And Reproducibility:**

This paper is one of the most approachable and significant papers I've reviewed.

**Strength And Weaknesses:**

Given that the use of offline reinforcement learning, as is, remains problematic, this paper offers a both sound and practical way to put it to use. It's refreshing to see an idea with apparent merit demonstrated to the point where it can be implemented, And bringing the user back, to propose a user centric approach that touches on the "representation" theme of the conference is notable.

I wish the authors had mentioned the computational effort needed for training.  Another thought is to be more specific about how choices are presented to the user -- do they chose only \lambda, or do they also get to see the policy that their choice implies?  One might imagine that exposing the risk / reward tradeoff implicit in the choice, in terms of the reward distribution would be  useful. None of these wishes are flaws, but rather extensions should the authors wish to address them.

**Summary Of The Paper:**

This paper takes a user-centered approach to the problem of the behavior of off-line reinforcement learning models when put into use. It takes an idea that is simple in concept and practical to apply that addresses some of the risks of using RL models that have been trained off-line. The idea is that users be given a choice of how aggressive or conservative a policy they follow by parameterizing the choice as a single parameter learnt at training time.  Learning this parameter has a clear computation solution, and the method can be demonstrated to generate a smooth transition from a given behavioral policy to a more aggressive one.

**Summary Of The Review:**

By considering a user-centric approach, this paper addresses an outstanding dilemma about implementing off-line RL policies in a way that is easily representable to the user and builds on current offline methods.

---

### Decision · Program_Chairs · 2023-01-20

**Decision:**

Accept: poster

**Justification For Why Not Higher Score:**

It will be nice if experiments can also be performed on classical benchmarks such as D4RL, so that comparison can be made with state of the art.

**Justification For Why Not Lower Score:**

The problem and method are very practical, and can benefit many applications.  The empirical performance also appears encouraging.

**Metareview: Summary, Strengths And Weaknesses:**

This paper proposes a new approach to address the out-of-distribution issue in offline-to-online reinforcement learning.  It allows uses to tune at runtime the budget of deviation from the original policy, so that the agent can start by being conservative, but gradually become aggressive until the policy or behavior needs to be limited.

The problem and method are very practical, and can benefit many applications.  The empirical performance also appears encouraging.  It will be nice if experiments can also be performed on classical benchmarks such as D4RL, so that comparison can be made with state of the art.


**Note From Pc:**

if the above contains the word "oral" or "spotlight" please see: "oral" presentation means -> notable-top-5% and "spotlight" means -> notable-top-25%. As stated in our emails, we are disassociating presentation type from AC recommendations